# Reducing the Incidence of Skin Cancer through Landscape Architecture Design Education

**Wendy McWilliam** [1],*, **Andreas Wesener** [1], **Anupriya Sukumar** [1] and **Robert D. Brown** [2]

1    School of Landscape Architecture, Faculty of Environment, Society and Design, Lincoln University, Lincoln 7647, New Zealand; andreas.wesener@lincoln.ac.nz (A.W.); anupriya.sukumar@lincolnuni.ac.nz (A.S.)
2    Department of Landscape Architecture and Urban Planning, Texas A&M University, College Station, TX 77840, USA; rbrown@arch.tamu.edu
*    Correspondence: wendy.mcwilliam@lincoln.ac.nz; Tel.: +64-3-423-0477

**Abstract:** There is a high and growing incidence of skin cancer associated with overexposure to the sun. Most of a person's exposure occurs during their first eighteen years of life. While many children are taught to wear hats and sunscreen, studies indicate these are inadequate. There is a pressing need to improve the design of our landscapes to reduce exposure. Landscape architects can play a key role in driving this process, but only if they understand the factors determining sun protection behaviours among children in the landscape, and how to design for these. We introduced a systematic evidence-based teaching approach to landscape architecture students in New Zealand where the incidence of skin cancer is one of the highest in the world. In this paper, we describe the methods we used to integrate scientific information into a creative design process that included four design phases: (1) review, summary and translation of evidential theory into design guidelines; (2) inventory and analysis of existing schoolyard; (3) redesign of schoolyard; and (4) final design evaluation. We found this process was effective in developing student appreciation for the need to improve sun protection through design, for increasing their understanding of the evidential science, in addition to developing their ability to translate, often inaccessible, evidential data into its spatial form implications. Furthermore, the process led to a high degree of confidence and pride among many students as their resulting design solutions were not only supported by evidence but were often highly creative. Such evidence-based design courses are essential for preparing future landscape architects to design landscapes that significantly reduce the incidence and health effects of skin cancer.

**Keywords:** UVR exposure; sun protection; schoolyard design; evidence-based design; evidence-based landscape architecture; research through designing; southern hemisphere; New Zealand

## 1. Introduction

Overexposure to ultraviolet radiation (UVR) from the sun leads to multiple negative health effects [1,2], that are anticipated to worsen with increased UVR exposure associated with climate change [3]. Health effects include sunburn (erythema), skin aging, increased frequency of illnesses and vulnerability to disease [4–8], suppression of the immune system, cataracts [9,10], and skin cancer [11–13]. New Zealand has one of the highest incidences of both melanoma and non-melanoma skin cancer in the world [14,15]. Non-melanoma skin cancer is the most commonly occurring. Globally, its incidence is rapidly increasing among fair-skinned people [16,17]. In New Zealand, it affects about 100,000 each year [18]. While the non-melanoma type is most commonly reported worldwide [19], and does not lead to loss of life [20] it can impose a considerable economic burden on the health-care system [21]. The incidence of melanoma is equally significant in New Zealand, with a mortality rate

significantly higher than that associated with road accidents [22]. Furthermore, the health care costs associated with melanoma are estimated at more than NZD 24 million per annum [23].

The geographical location of a country strongly influences the amount of UVR exposure. Regions located in mid-southern latitudes experience significantly higher UV levels compared to the mid-northern latitudes due to the reduced earth–sun distance in summer, reduced stratospheric ozone (also known as the 'ozone hole') and unpolluted skies [24]. New Zealand, located between 35° S and 46° S latitude [25], is isolated from other large landmasses and receives a constant flow of essentially unpolluted air from a westerly air-stream. These factors results in 25% and 40% higher UV levels than North America [26] and Europe [27,28]. Epidemiological studies indicate that outdoor recreational activities such as skiing or mountaineering are associated with an increased risk of skin cancer [29]. Many New Zealanders value outdoor lifestyles and tend to spend significant portions of their time outdoors [30]. Much of it occurs during peak UVR exposure times [24] due to the country's relatively cool summers. In addition, New Zealand has a large proportion of European-descent population with lighter skin colour, which makes them generally vulnerable to skin cancer [30,31]. Light skin has less UVR reflectance ability [32], and light-skinned individuals have a higher risk of developing skin cancer. With climate change, the incidence of UV-B radiation is expected to increase in New Zealand [3] including the number of extreme (hot) temperature days [32]. The World Health Organization recommends wearing protective clothing, sunglasses and the application of sunscreen, in addition to designing landscapes to reduce the amount of UVR through shade provision [33].

A majority of a person's lifetime UVR exposure occurs within their first 18 years of life [34,35]. Overexposure during these years significantly increases the risk of developing skin cancer later in life [3]. The World Health Organization [36] recommends primary schools as priority settings for skin cancer prevention efforts, as they allow access to young children up to 13 years of age when such behaviours can be most effectively influenced. They also allow access to other stakeholders, such as teachers and parents, who are key influencers of this age group [37]. In addition, this age group has been targeted because they are particularly vulnerable to overexposure. The dermis layer of the skin of children aged 2 to 13 is significantly thinner than that of adults, and therefore more easily damaged by UVR [38]. Furthermore, children spend a significant portion of their day outdoors. In New Zealand, school children aged between 8 and 12 years, on average, spend 2.3 h/day outdoors between 7:00 and 18:00 h, and receive three-quarters of their total daily UVR exposure during this time [39].

The SunSmart programme is a school-based health promotion programme originally developed by the Cancer Council Victoria (Australia) in 1993. It employs 'accreditation' as a process of implementing sun protection programmes in schools [40,41]. The Cancer Society New Zealand (CSNZ) adapted the programme to New Zealand conditions. It was launched nation-wide among primary schools in 2005 [42] and adopted the WHO [43] framework including (a) sun protection education; (b) healthy school environment (social and physical); and (c) community participation to reduce the incidence of skin cancer among this age group. To implement sun-safety practices in primary schools, the CSNZ developed guidelines, including sun-protection policies and criteria recommended for effective implementation [44]. SunSmart accreditation was offered to participating public schools, who were to implement measures to encourage sun protection behaviours among children including the wearing of hats, and the use of sunscreen. Additionally, the CSNZ developed shade design guidelines for organisations (i.e., schools, offices, local authorities, sporting venues), professionals, and individuals involved in shade planning and design [45]. Accredited schools were asked to provide adequate shade in playgrounds. However, the programme guidelines do not provide specific information on how to design playgrounds to reduce exposure to indirect UVR, for example through the incorporation of less reflective surface materials. Evaluations of these programmes indicated an inconsistency towards the implementation of sun-safety practices [46], suggesting that primary schools have implemented only some of the programme criteria. While programmes have been successful in increasing the number of children wearing hats and using sunscreen [47–50], they were less successful in reducing UVR exposure through the design of schoolyards [49]. For sun protection behaviours to develop among

children, school landscapes need to be designed to provide protection from both direct and indirect UVR [51,52], attractive activities, as well as comfortable microclimates [53]. Playgrounds sometimes lacked sufficient shade cover and used large areas of reflective materials. In addition, where shade was provided, children were not playing, either due to a lack of attractive activities or poor thermal comfort (i.e., shaded areas were uncomfortably windy or cold).

Landscape architecture can play a key role in designing schoolyards with improved attributes for sun protection. There is well developed scholarly literature on designing attractive play spaces in support of healthy child development [54,55]. There are also bodies of theory around modifying microclimate through design [53]. However, encouraging sun protection behaviour design has yet to be effectively addressed in landscape architecture teaching, and is badly needed if we are to protect our populations against the multiple and serious health impacts associated with UVR overexposure. For effective design solutions, the design of our landscapes needs to be based upon research [56,57]. Furthermore, there is a need to test alternative solutions through design [58], and subsequently evaluate implemented designs for their support for anticipated sun protection behaviours among children.

## 2. Methodology

In this paper, we explore a 'research through designing' (RTD) methodology [59] to teach design in support of sun protection behaviours to undergraduate landscape architecture students. The teaching method focuses on the development and application of evidence-based design guidelines in support of sun protection behaviours [60] in an academic studio-based learning and teaching environment. Within the (post)positivist research paradigm, RTD supports the production of empirical, evidence-based, generalizable specialist knowledge that "can take the form of generally applicable insights and design guidelines" [59] (p. 122). Evidence-based design (EBD) and the related notion of evidence-based landscape architecture (EBLA) as "the deliberate and explicit use of scholarly evidence in making decisions about the use and shaping of land" [56] (p. 328) has gained increasing attention in recent years. A recent project review illustrates how landscape architects who applied EBLA methods were able to achieve more successful design outcomes than with 'conventional' design approaches [57]. The identified need for advancing EBLA as an integrative approach for practicing landscape architects to produce better design outcomes creates new responsibilities for landscape architecture educators [56].

The paper discusses a RTD teaching and learning approach at the School of Landscape Architecture (SOLA) at Lincoln University, New Zealand. The approach was followed over three years, during the period 2018–2020, in a third-year studio "Sustainable Design and Planning" (LASC 322), a core curriculum course within SOLA's four-year Bachelor of Landscape Architecture (BLA) programme. Landscape architecture undergraduate students were required to improve the design of a schoolyard, located in the inland South Island region of New Zealand, in terms of sun protection for children. According to the Köppen–Geiger climate classification system, this region has a Cfb, or marine mild winter climate with a cool summer, and cool, but not cold winter. The SOLA Studio projects replicate 'real life' practice-oriented design challenges where students solve real problems associated with landscapes in their community. This is a learner-centred approach [61,62] where students are encouraged to identify and explore problems and solutions themselves with support from tutors. This approach to teaching has been found to result in improved learning outcomes as it increases the project's relevance among students and their motivation to solve design problems [63]. The studio project was eight weeks in duration including up to 10 tutorial hours per week in 2018. It was extended to twelve weeks in 2019 and 2020 in alignment with the approach of Lenzholzer and Brown [58].

There is no didactic literature in Landscape Architecture on landscape design in support of sun protection for children. Therefore, we initially followed Lenzholzer and Brown's teaching approach for climate responsive design [58]. Their approach involved three main steps (1) review and summary of evidential theory in the scientific literature in support of the design problem, (2) inventory and analysis of a landscape in terms of how well its design is supported by the scientific literature,

in terms of solving the design problem, and (3) the development of alternative designs to improve the manner in which the landscape solves the design problem. Over the course of three years, we revised their approach according to our teaching experience into four steps: (1) the review, summary and translation of evidential theory into design guidelines, (2) inventorying and analysing of existing landscape with respect to design guidelines, (3) redesign of landscape in support of design guidelines, and (4) evaluation of redesign relative to existing design, with respect to design guidelines and best management practice. The steps are presented here sequentially, but in practice are often implemented by students cyclically, or in accordance with their needs.

Students were required to submit three outputs for the assessment of their performance. The first assessment required students to submit design guidelines at three spatial scales—component, activity zone (involving multiple components determining sun protection), and schoolyard (involving multiple activity zones) in support of protecting children in schoolyards from skin cancer. The second assessment required the submission of an inventory and analysis of an existing schoolyard study site with respect to the extent to which it met their guidelines. Lastly, students submitted a minimum of three drawings showing the re-design of the existing schoolyard to better meet their design guidelines. This included an evaluation of the extent to which their re-designed schoolyard better met their design guidelines relative to the existing schoolyard design.

## 3. Results

### 3.1. Review, Summary and Translation of Evidential Theory into Design Guidelines

We dedicated the first four weeks to the review and summary of evidential theory in support of design that protects children from skin cancer in schoolyards. In previous studios, this step was integrated into the inventory and analysis phase of the design process. It was not given much importance in assessments. However, we found that students did not have sufficient time or motivation to adequately find the evidence in support of their designs.

We started this project by providing a lecture that introduced the design problem to students, supported by a design brief. We described the design problem to be addressed, the EBD process for solving it, and the three assessments of student performance. In past studios, we asked students to develop their own design goals based on those given by their "client" in the brief. However, we found that when students defined their own goals, they often developed too many, not realizing that to meet each goal based on evidence would require them to review and summarize an often different body of literature. Most students lacked the time and motivation needed to conduct multiple literature reviews in support of design solutions. In order to overcome this barrier, we provided students, initially, with one main design goal: to improve sun protection behaviour among children in schoolyards. We also provided a tutorial that demonstrated how to translate goals into relevant, clearly expressed, specific, and if possible, measurable objectives toward goal achievement. We followed this with students working in small groups, and then as a class, to practice developing objectives with these characteristics. We guided students in this process based on our literature review that identified four key factors to determine sun protection in schoolyards: (1) protection from direct UVR exposure; (2) protection from indirect UVR exposure; (3) thermal comfort; and (4) desirable passive and active schoolyard activities. Students developed these factors into four main design objectives that became key drivers of their EBD process.

The majority of our undergraduate students come into the studio with little background knowledge on most of the problems they are required to tackle. Landscape architects are usually generalists, rather than specialists, required to solve multiple problems in the landscape. This means that many students find it difficult to acquire specialised knowledge and find corresponding evidence in support of their design goals and objectives. To address this barrier, we gave them key sources of literature on the four key factors determining protection. This approach is in line with that of Lenzholzer and Brown [58] who provided selective readings to their undergraduate students rather than requiring them

to find them independently. We also synthesized information about the four key factors in a five-page summary document based on key readings. We explained to the students how the information was critically reviewed and synthesized so that they had a process to follow in their own reviews of the literature. We found this initial 'leg up' in the literature review process helped our students get started.

The literature supporting design problems is often scattered across multiple fields, and is often incomplete, with a degree of uncertainty. This means students find locating, synthesizing and summarizing the evidence time-consuming, difficult and uncomfortable. We could do little about the uncertainty inherent in the scientific evidence, other than to tell students to expect it and accept it. Landscape architects are frequently asked to solve spatial problems for which there is incomplete knowledge. We chose a design problem that we thought not too complex—as the greater the complexity of the problem, the more bodies of literature are required to provide sufficient evidence in support of solutions. Designing for sun protection required students to synthesize evidence about four variables determining protection, and many found this to be sufficiently challenging.

A further barrier to students accessing the literature in support of evidence was that it often contains jargon and formulae that students are unfamiliar with and this can make it incomprehensible to students [58]. To overcome this barrier, we found that we had to have expertise in the area of design to help students to comprehend what they were reading. We therefore decided to only take on design problems in our EBD studio for which we had conducted research, and therefore had a good knowledge of the literature.

A third barrier to students identifying evidence in support of their designs was that it is often not in a form that students can readily apply to solve spatial problems. Bereiter and Scardamalia [64] termed this type of information, "inert knowledge", or knowledge that students possess, but cannot access or use. For our students, this meant they had to guess what the spatial implications for their design might be, but without evidence to back up their assertions. We helped them to overcome this barrier in two ways. Firstly, we required students to develop design guidelines, using a generic schoolyard to illustrate their application. Design guidelines provide design principles or strategies (rather than specific solutions) to a design goal or objective. They suggest particular directions for design that are "neither totally specific nor completely universal and represent structured knowledge bundles at an intermediate level" [65] (chapter 12; p. 3). Their abstract but directed epistemological foundation makes them potentially transferable to a range of applications. Hence, design guidelines need to be based on research or scientific evidence, and require students to clearly define terms, such as active, passive, and transitional schoolyard activities. However, they also require the translation of this evidence into spatial implications which are essential for landscape architects to apply theory to solve design problems [60].

To assist them in this process, we gave a lecture on the attributes of effective design guidelines based on Evans [66], and graded student guidelines based on these attributes. Evans [66] found that effective guidelines had the following attributes: clear goals and objectives, theoretical principles and strategies, supporting evidence, supporting diagrams and images, concise descriptive text, open-ended guidelines, and measurable data. With these criteria in mind, we initially asked students to develop one guideline each from the summary of evidence we provided. Students then presented their guideline and we critiqued them as a group relative to the effective guideline criteria.

To assist students in providing and illustrating evidence in support of their guidelines (particularly where it was missing in the literature), we taught students to use the SketchUp 2017 software, to create a virtual 3D model, and use the geo-location and shadow tools. Students were asked to create a virtual base model of a generic schoolyard that we provided as a hand drawing. SketchUp allows students to locate a landscape in a specific location, which is vital for exploring microclimatic design. In this case, students located their generic schoolyards in the inland South Island region of New Zealand to determine the implications of their design guidelines for shade provision throughout the day and year.

Most students successfully developed comprehensive and professional looking guidelines. Figure 1 provides an example of a table of contents demonstrating that students provided information about the

science behind UVR overexposure, the need for sun protection in schoolyards, and design guidelines for protecting children from direct and indirect UVR exposure, thermally comfortable spaces and schoolyard activities in support of the health and wellbeing of children. The guidelines covered three spatial scales (components, activity zones and school wide), and addressed changing microclimates over the school year and at three key times of the day in terms of exposure, 10 a.m., noon, and 2 p.m. Sketchup 3D models were effectively used by students to generate evidence in support of their guidelines where it was missing in the literature.

## contents

**Figure 1.** Table of contents within guidelines for sun protection in schoolyards (Student: B. Thatcher 2019).

The guidelines allowed students to translate the science supporting sun protection into alternative spatial components for the protection of children from overexposure, and for their relative evaluation in support of sun protection (Figure 2).

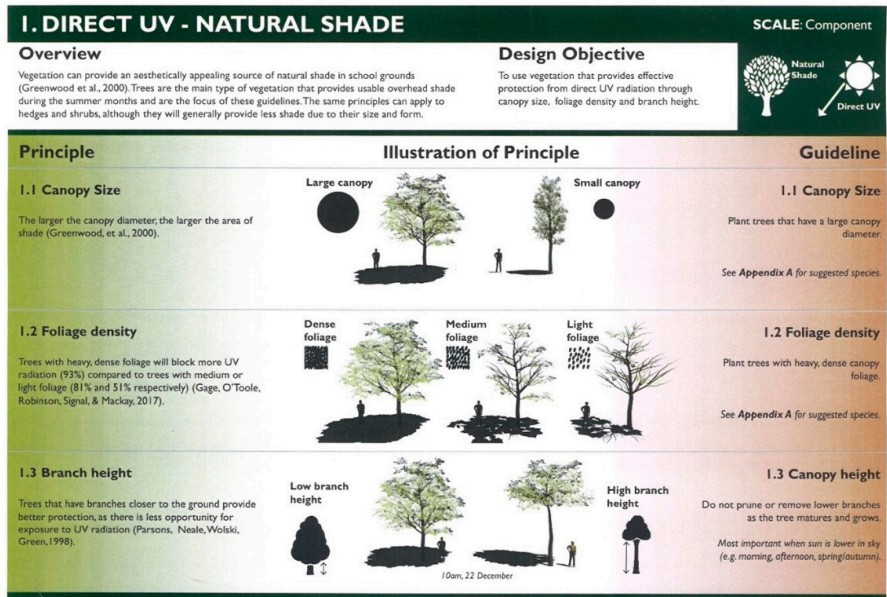

**Figure 2.** Component guideline for providing UVR protection with vegetation (Student: D. Pedley 2019).

The modelling also facilitated the process of combining the individual components into different activity zones (active zones, e.g., a sport field, passive zones, e.g., lunch area, and transition zones, e.g., entries and stairways). This helped students to explore alternative ways of supporting sun protection behaviours for all the different types of activities occurring within schoolyards (Figure 3).

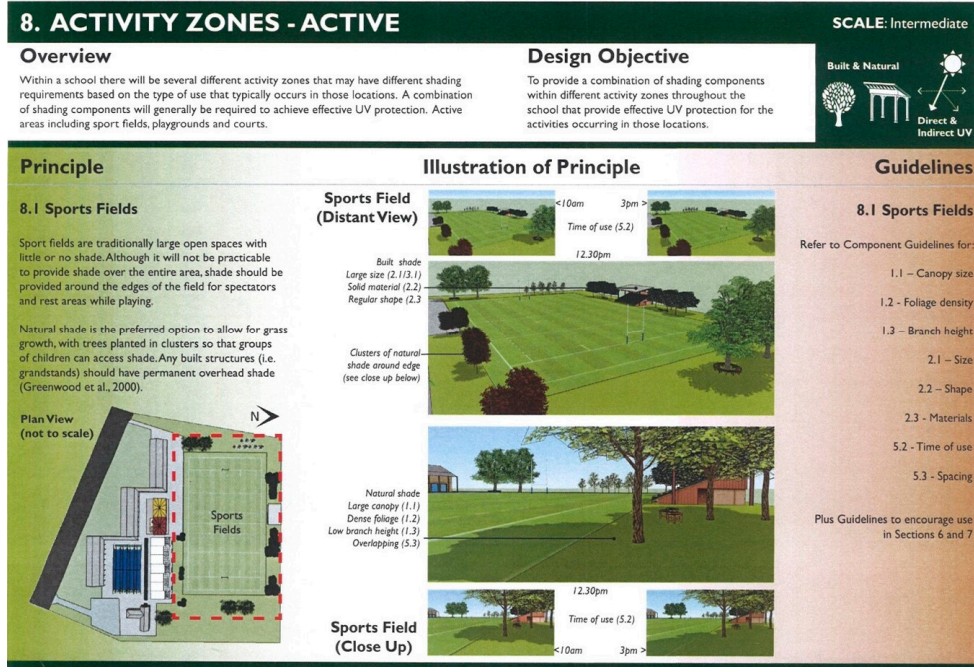

**Figure 3.** Activity zone guideline for an active zone activity, a play field (Student: D. Pedley 2019).

Finally, the students used the modelling to combine their sun-protected activities into an integrated schoolyard wide plan, looking at how the protection provided in one zone influenced another, and how adjacent land uses may affect sun protection within a schoolyard (Figure 4).

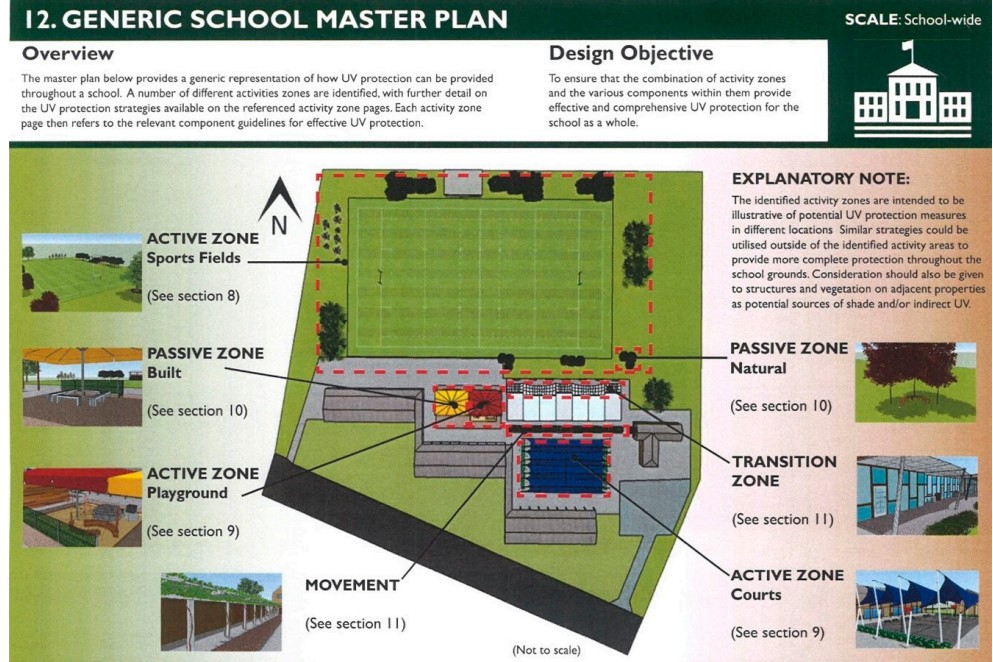

**Figure 4.** Schoolyard-wide guideline for combining activity zone types (Student: D. Pedley 2019).

### 3.2. Inventorying and Analysing of Existing Landscape with Respect to Design Guidelines

Having developed their guidelines and applied them to a generic schoolyard, we then asked students to use them to redesign an existing public schoolyard in Christchurch. We focused student attention on the four key factors required to meet their goal of maximizing sun protection: direct UVR exposure; indirect UVR exposure; thermal comfort; desirable activities. We gave a tutorial, followed by a workshop, on identifying what students would need to inventory in the schoolyard to determine the extent to which the existing schoolyard met these objectives. We introduced a simple chart to assist students to identify these factors and their relationship with their objectives and goal (Figure 5).

| Goal | Objectives | Inventory |
|------|-----------|-----------|
| **1.** | 1.1 | |
| | | |
| | 1.2 | |
| | | |
| | 1.3 | |
| | | |

**Figure 5.** Simple chart helping students ensure design goals, objectives and inventory are clearly related.

Following this, we visited the schoolyard and students worked in teams to describe, map, photograph and where relevant, draw their inventory. After sharing their inventory with others in their group, they developed a SketchUp 3D model of the schoolyard, including all inventory elements determining sun protection. We then asked students to analyse the schoolyard for its support for sun protection. During 2018 and 2019, this resulted in most students providing an analysis of the extent to which each of the four key design objectives were met independently (Figure 6).

While the inventory and analysis process allowed students to determine where factors affecting protection were and were not in place, there was insufficient evidence in the literature to help students evaluate the extent to which their inventoried elements protected children from overexposure to the sun. For example, the models allowed them to see where shade was generated and whether it occupied a small or large proportion of an activity area; however, they did not help students determine what percentage of shade was adequate or inadequate. This type of information was largely missing in the literature. Therefore, to assist the students with their analysis, in 2020, we established assumptions and targets for protection based on best evidence in the literature. For example, for each activity type, we developed minimum percentage shade cover requirements for protection. These were also used as target shade levels for the redesign, and, furthermore, for student evaluations of their success in meeting their sun protection goal.

Lastly, in 2020, we taught students the process of defining landscape units in support of design objectives, as described by Brown [67], in order to analyse the schoolyard for all four factors together rather than separately. Site capability for protecting children from skin cancer depends on all four factors being in place in each activity area. The simplification of each factor into unit areas, and their combination into joint units allowed students to clearly see where protection attributes were present and missing (Figure 7). However, we found that the process of defining units was too complex for many students to grasp quickly.

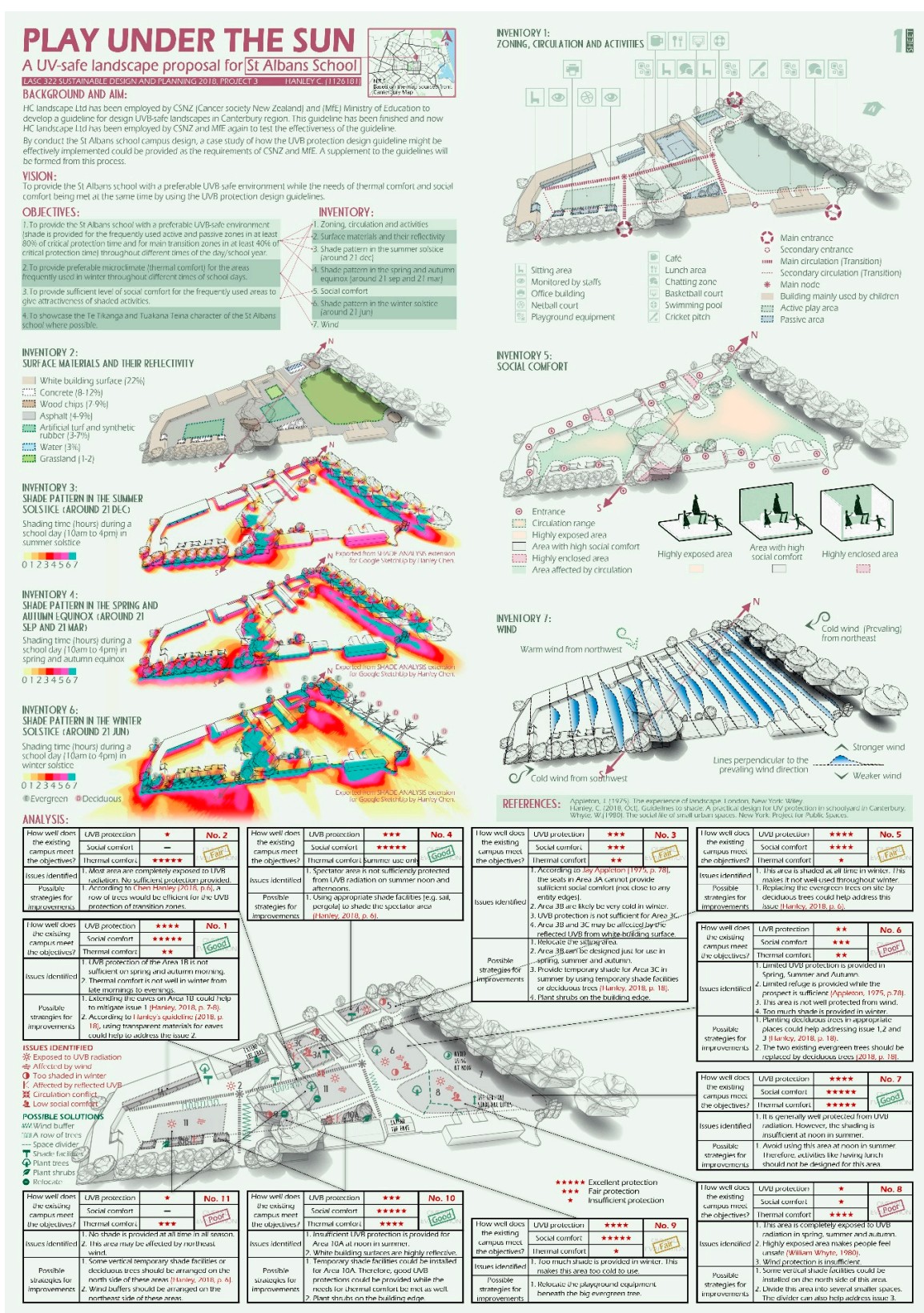

**Figure 6.** Inventory and analysis for sun protection by individual factors determining sun protection (Student: H. Chen 2018).

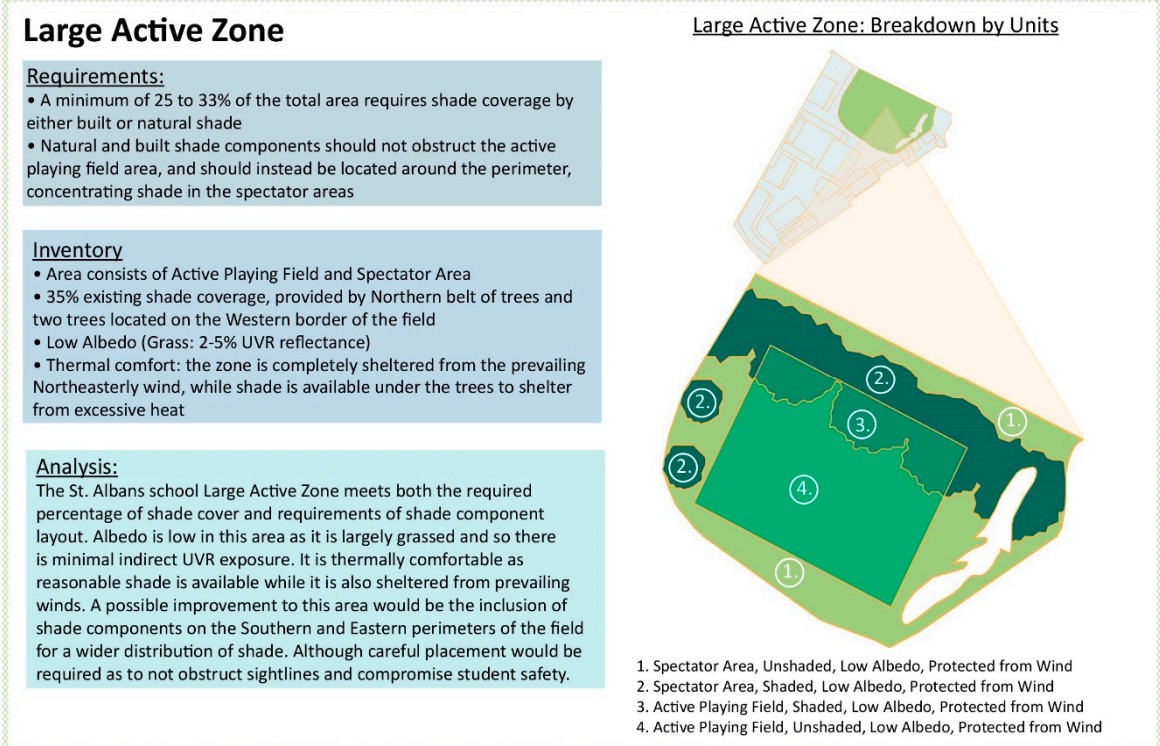

**Figure 7.** Drawing of landscape units (right) for protecting children and spectators from the sun in a large active zone (a playfield). The boundaries of the activity zone are determined by all areas associated with the activity including spectator areas. The dark green strip along the top of the drawing is the shade provided by a row of trees. The medium green is the mown grass playfield area. (Student: N. Kent 2020).

### 3.3. Redesign of Landscape in Support of Design Guidelines

At this point in their EBD design process, students were ready to shift their thinking from literature review and focused problem solving to freer thinking creativity. Like Lenzholzer and Brown (2013), we used creative exercises to encourage the development of a wide range of ideas and innovation inspired by their understanding of design in support of sun protection. Many students chose to create new activities and learning opportunities, rather than just improve the existing design in terms of providing more shade, changing materials to reduce reflectivity, or improving thermal comfort. Importantly, these students turned to the literature to further develop, evaluate, and provide supporting evidence for their ideas. This demonstrated they understood and could apply the EBD process to meet a new design objective (Figure 8).

### 3.4. Final Design Evaluation

While completed designs, communicated largely through modelling, appeared to better protect children from excessive sun exposure (particularly with respect to percent shade cover), we formalized this evaluation by asking students to repeat key aspects of their inventory and analysis process to determine the extent to which the schoolyard designs met each of the four design objectives in support of sun protection (Figure 9). In addition, in 2020, the better students measured the extent to which each activity type met their performance targets, and a few repeated their definition and evaluation of landscape units in their redesigned schoolyard to determine the extent to which they had met all four of their design objectives in each activity zone.

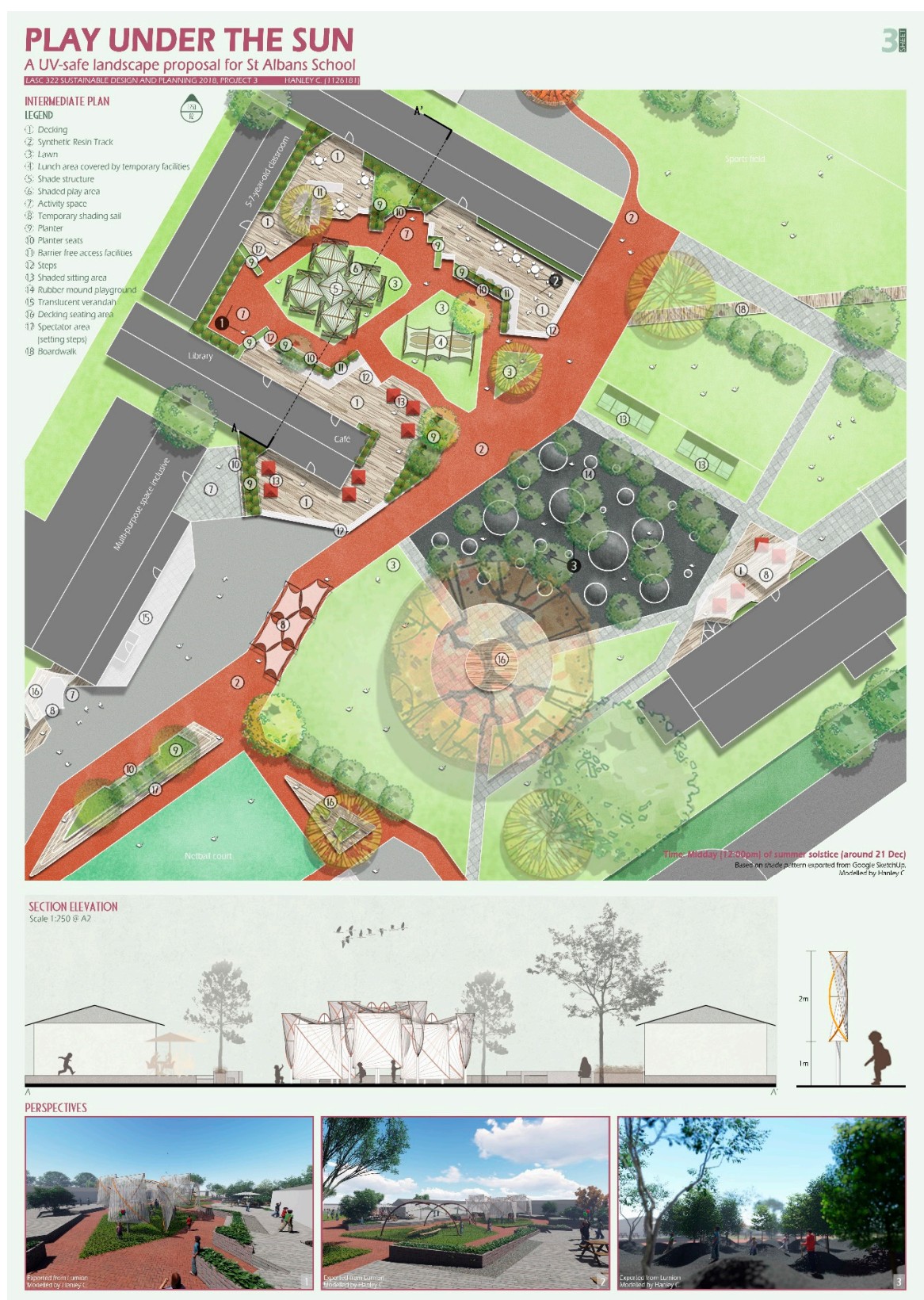

**Figure 8.** Redesigned playground with improved support for sun protection (Student: H. Chen 2018).

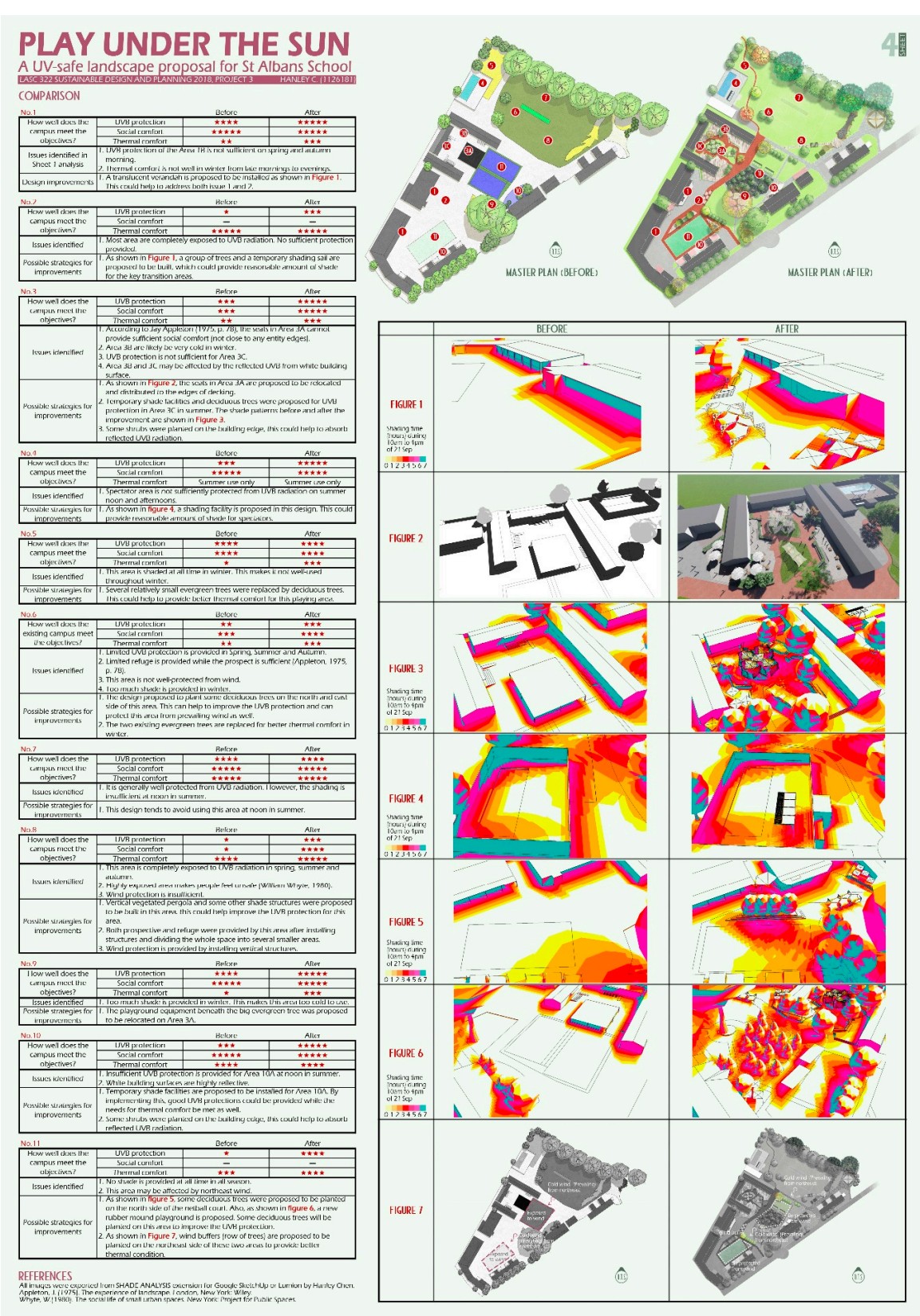

**Figure 9.** Evaluation of the extent to which sun protection design objectives are met in redesigned playground relative to existing. Colours in Figures 1–6 indicate the number of hours of shading (Student: H. Chen 2018).

## 4. Discussion

We built on the experiences of Lenzholzer and Brown [58] in teaching our EBD approach to overcome some of the didactic problems they identified. We found that most of these barriers to EBD occurred in the first two phases of the process. Success in these two phases was essential for success in the latter phases. Scholars generally agree that familiarity with the subject matter is essential for higher level thinking and problem solving skills to develop [68]. However, few Landscape Architecture students begin their design process with more than superficial knowledge or comprehension of the subject matter needed to solve design problems. We found that undergraduate students (and especially those with English as a second language) had difficulty understanding and summarizing the evidence from the scientific literature, particularly when evidence came from different subject fields, each with their own terms and ways in which they communicate results. More experienced students, and particularly those arriving with a prior university degree, were in general more able to access the literature. Generally, they enjoyed the first two phases of the EBD process which required well developed critical thinking, and English language reading, comprehension, and writing skills. However, with more time and support, in terms of providing an initial summary of the literature, the provision of key papers, and tutorials on how to access and translate the literature to address a design problem, younger undergraduate students were able to successfully pass through these phases.

The translation of the evidence into spatial implications through the preparation of design guidelines, supported by 3D modelling, was successful in helping students to transform inert knowledge into spatial forms. However, while this particular design project lends itself to exploration through SketchUp, this software may not be applicable to all, particularly coarser scaled, design problems (although other more appropriate software may be available to support their exploration).

The 3D modelling was also a powerful tool for students to facilitate their inventory and analysis in phase two. Students were able to develop highly convincing realistic models of how each variable expressed itself in the playground in support of, or against, sun protection. This enabled students to analyse how each element determining behaviour manifested on the playground and needed to be addressed. The establishment of shade performance targets helped students to make decisions about the significance of what they analysed, and the extent to which they needed to improve shading. Similar targets for the other variables may also prove useful in helping to guide and stretch students toward improved protection. Students who went further in this phase to analyse their sites for landscape units of shared variables determining sun protection, told us that it cemented their understanding of how the variables came together to determine protection. However, relatively few students advanced to this level of sophisticated analysis. Only one week was dedicated to teaching this process, and for many students, this was an inadequate amount of time. An additional week, and dedicated assessment, might improve its uptake.

Feedback from students over the three years, both during and following studio completion, indicated that the applied RTD teaching and learning approach and related EBD processes helped students gain a deeper understanding and appreciation for protecting children from overexposure from the sun. In addition, students felt confident they knew how to repeat the EBD process in future practice to improve landscape design in support of reduced cancer incidence among children. Students said they acquired a feeling of confidence in their design solutions they had not experienced in other design studios, as they were able to support their solutions with evidence and could demonstrate through modelling, and through the measurement of their performance targets, that they had indeed improved sun protection for children in their schoolyards. A further measurement of success was the interest in the work expressed by stakeholders, such as the New Zealand Cancer Society, who recognized that many schoolyards are designed without adequate protection from the sun, and are looking for ways to improve their design.

While the EBD process is sometimes criticised for being overly based on existing knowledge, and not enough on the generation of new ideas or innovation, we did not find this to be the case. On the contrary, the evidence identified and translated into spatial form served as a basis or infrastructure upon

which innovations in design could be developed and tested. However, resulting 'design hypotheses' still require implementation, monitoring and refinement in an adaptive EBD process to ensure they are effective in protecting children from skin cancer.

## 5. Conclusions

The high and growing incidence of skin cancer will increasingly focus attention on finding ways to reduce its incidence within many countries, particularly during the first 18 years of life when most of a person's lifetime UVR exposure and damage occurs. Hats and sunscreen are not sufficient to protect our children. We need landscape design that not only provides wonderful, educational, and thermally comfortable areas for children to play, but protection from over exposure to the sun. More schools need to educate their students on designing for this important issue using an evidence-based design approach.

We found that while Lenzholzer and Brown's evidence-based design (EBD) teaching approach [59] provided a good foundation upon which to build our EBD teaching methodology, improvements were needed to overcome barriers to learning, particularly for many of our younger, less experienced, undergraduate students. These improvements increased student access to the evidential theory, and their success in translating it to spatial forms. In addition to providing students with sufficient time and motivation (through focused assessments), we recommend that examiners attempting similar studios incorporate the following teaching innovations into their teaching methodology.

Firstly, teachers need to provide tutorials to improve access to the evidential literature. Tutorials should include how to identify key publications, and how to critically identify and synthesize relevant theory from those publications.

Secondly, students need to be made aware that design goals and objectives determine the evidence needed to support their design. Furthermore, the inventory data they collect should help them evaluate the extent to which existing and proposed design interventions meet their goals and objectives. A limited number of goals, and specific (unambiguous), clearly communicated, and, if possible, measureable design objectives are needed to drive the EBD process. Without these, students are rudderless. To assist in this, we also recommend that targets be made, in concert with students. For example, the percentage shade targets per activity type we developed helped students determine the extent to which each activity space protected children from direct exposure to the sun before and after their redesign. Such targets would also be useful in more advanced EBD studios with graduate students who may be conducting research or post occupancy assessments to determine the extent to which a "design hypothesis" was successful in supporting sun protection.

Thirdly, students need support in translating non-spatial theory that may be in the form of words or numbers in the literature, into useable spatial form. Otherwise, the evidence they uncover is useless, and students have to "guess" the implications, resulting in designs unlikely to support objectives. We found the development of design guidelines as an EBD step highly effective in teaching students the relationship between theory and spatial design. Guidelines provided students with support throughout the whole EBD process. When students were unsure whether a design idea was effective, we could refer them to their own guidelines for guidance.

Fourthly, where the evidence was missing, with respect to the impact of natural and artificial structures and activities on sun protection, the 3D model was a key supporting tool to facilitate EBD. Such tools; however, often require a dedicated workshop to teach students how to use them. This also needs to be incorporated into the teaching programme. SketchUp is useful for certain design problems; however, further research is needed to identify other, possibly interdisciplinary, tools available in support other EBD design goals and objectives.

Finally, effective and simple ways to explore and teach design problems involving multiple variables need to be identified, or refined. Brown's landscape unit method [67] of analysing space in support of more than one variable has potential to fill this gap for problems involving only a few variables. It is relatively simple and does not require students to learn additional software;

however, more than a week is required to teach this method effectively to many undergraduate students. For problems involving many variables, this method is less effective as the greater the number, the more complex and difficult it is for students to identify units and understand their design implications. GIS analysis tools are likely to be required for their exploration; however, teaching GIS requires more time than is available in a one semester design studio, and is best taught in a separate, but related course.

**Author Contributions:** Conceptualization, methodology, validation, formal analysis, investigation W.M. and A.W.; resources, data curation W.M., A.W. and A.S.; writing—original draft preparation W.M. and A.W.; writing—review and editing W.M., A.W., A.S. and R.D.B.; supervision, project administration W.M. and A.W. All authors have read and agreed to the published version of the manuscript.

**Funding:** This research received no external funding.

**Acknowledgments:** We would like to thank all our tutors who helped us teach this course between 2018 and 2020, and our students. Without them this research would not have been possible. We would also like to thank St. Albans School in Christchurch and the New Zealand Cancer Society for their support.

**Conflicts of Interest:** The authors declare no conflict of interest.

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
