# Peer review of "Reducing the Incidence of Skin Cancer through Landscape Architecture Design Education"

_sustainability, doi:10.3390/su12229402_

Round 1

Reviewer 1 Report

A very good article with a rigorous methodology. i greatly appreciate the choice of authors to facilitate student engagement and to empower them in addressing a very relevant topic for personal and social health, for sustainability, and for education.

Reviewer 2 Report

Overall, well-written paper with clearly defined goals and outcomes supporting Evidence Based Design.  In the introduction authors state that melanoma is not fatal but has health and economic consequences.  What are they and how do the proposed outcomes alleviate the negative effects of UV exposure? What are the advantages or disadvantages of using natural shade-giving landscape features, such as trees, versus man-made elements, such as pergolas?